# *Salvia plebeia* R. Br. Water Extract Ameliorates Hepatic Steatosis in a Non-Alcoholic Fatty Liver Disease Model by Regulating the AMPK Pathway

**DOI:** 10.3390/nu14245379

**Published:** 2022-12-18

**Authors:** Subin Bae, Yoo-Hyun Lee, Jeongmin Lee, Jeongjin Park, Woojin Jun

**Affiliations:** 1Division of Food and Nutrition, Chonnam National University, Gwangju 61186, Republic of Korea; 2Department of Food and Nutrition, The University of Suwon, Hwaseong 18323, Republic of Korea; 3Department of Medical Nutrition, Kyung Hee University, Yongin 17104, Republic of Korea

**Keywords:** *Salvia plebeia* R. Br., non-alcoholic fatty liver disease, high-fat diet, steatosis, AMPK pathway

## Abstract

*Salvia plebeia* R. Br. (SP), grown from autumn to spring, is used as a medicinal herb from roots to leaves. This herb exhibits antioxidant activities and various physiological effect, including anti-asthma, immune-promoting, anti-obesity, and anti-cholesterol effects. However, the effectiveness of SP against non-alcoholic fatty liver disease (NAFLD) and the associated mechanism have not been elucidated. In this study, alleviation of NAFLD by SP was confirmed in a mouse model of hepatic steatosis induced by a high-fat diet and in HepG2 cells administered free fatty acids (FFA). In the experimental model, intrahepatic lipid accumulation was investigated using the AdipoRed^TM^ assay, Oil Red O staining, biomarker analysis, and hematoxylin and eosin staining. Furthermore, glucose tolerance was examined based on the fasting glucose levels and oral glucose tolerance. The molecular mechanisms related to hepatic steatosis were determined based on marker mRNA levels. Blood FFAs were found to flow into the liver via the action of fatty acid translocase, cluster of differentiation 36, and fatty acid transporter proteins 2 and 5. *Salvia plebeia* R. Br. water extract (SPW) suppressed the FFAs inflow by regulating the expression of the above-mentioned proteins. Notably, modulating the expression of AMP-activated protein kinase (AMPK) and liver X receptor, which are involved in the regulation of lipid metabolism, stimulated peroxisome proliferator activated receptor α in the nucleus to induce the expression genes involved in β-oxidation and increase β-oxidation in the mitochondria. AMPK modulation also increased the expression of sterol regulatory element binding protein-1c, which activated lipid synthesis enzymes. As a consequence of these events, triglyceride synthesis was reduced and lipid accumulation in hepatocytes was alleviated. Overall, our findings suggested that SPW could ameliorate NAFLD by inhibiting hepatic steatosis through AMPK modulation.

## 1. Introduction

The global prevalence of non-alcoholic fatty liver disease (NAFLD) is approximately 25% and has been rapidly increasing over the past decade [1]. An increase in NAFLD is caused by overnutrition owing to industrialization and lifestyle changes. Further, these changes are considered to affect the prevalence of NAFLD by causing metabolic disorders [2]. NAFLD is recognized as an important factor of metabolic syndrome, in addition to obesity, diabetes, and dyslipidemia. In fact, this disorder is an independent risk factor for the development of various hepatic diseases, including cardiovascular disease, diabetes, chronic kidney disease, and malignancy. Moreover, NAFLD is known to be closely related to various factors, such as genetic and environmental factors and metabolic disorders. Accordingly, the focus of NAFLD is increasing.

NAFLD covers all conditions from simple steatosis, steatohepatitis, and cirrhosis to hepatic fibrosis [3]. The pathophysiology of NAFLD is related to the induction of lipid accumulation due to obesity, increased free fatty acid (FFA) levels, and the resulting increase in oxidative stress, mitochondria dysfunction, secretion of inflammatory cytokines, and apoptosis [3,4]. When various factors, such as overnutrition and high-fat diet (HFD), cause an increase in FFA levels in the body, the upregulation of cluster of differentiation 36 (CD36), fatty acid transport protein 2 (FATP2), and fatty acid transport protein 5 (FATP5), which are lipid transporters in the liver cell membrane, increases the influx of FFAs into the liver [5,6]. When the amount of FFAs flowing into the liver increases, FFAs are converted into triglyceride and begin to accumulate [3,7]. As the expression of AMP-activated protein kinase (AMPK) and liver X receptor (LXR) is up- and down-regulated, the expression of markers related to lipid synthesis, such as sterol regulatory element binding protein-1c (SREBP-1c), acetyl-CoA carboxylase (ACC), and fatty acid synthase (FAS) increases, while those of markers related to β-oxidation, peroxisome proliferator-activated receptors α (PPARα), and carnitine palmitoyl transferase-1α (CPT-1) decreases [8,9,10]. As a consequence, lipid accumulation increases and NAFLD develops. The elevated oxidative stress causes a deepening of ballooning in hepatocytes, cell death, and lipid peroxidation, which causes liver damage and suppresses the activation of hepatic stellate cells, thus leading to liver fibrosis. Therefore, to confirm the alleviation of NAFLD, the molecular signaling mechanisms that ameliorate lipid accumulation and suppress oxidative stress in the liver must be elucidated.

*Salvia plebeia* R. Br. (plebeian herba), whose fragrant branches are called “yeojicho”, is an annual or biennial herb belonging to the lamiaceae family that was distributed in many countries such as Korea, China, Japan, and Australia [11,12,13]. *Salvia plebeia* R. Br. is used as a folk medicine to treat several inflammatory diseases (including hepatitis), cough, diarrhea, tumors, and hemorrhoids in Korea [11,14]. This herb contains various bioactive compounds, including caffeic acid, rosmarinic acid, luteolin, nepetin, saponin, hispidulin, and homoplantaginin [15,16]. In previous studies, *Salvia plebeia* R. Br. water extract (SPW) was demonstrated to exhibit a variety of biological activities. In fact, SPW was found to exhibit effective antioxidative, anti-cholesterol, and anti-obesity activities. Accordingly, SPW is thought to be effective for inhibiting lipid accumulation [17,18]. However, no experimental studies have sought to determine the beneficial effects of *Salvia plebeia* R. Br. against NAFLD.

The current study aimed to investigate whether SPW alleviates hepatic steatosis. Overall, our findings provide new insights into the effect of SPW on NAFLD in FFA-induced HepG2 cells and HFD obese mice.

## 2. Materials and Methods

### 2.1. Preparation of SPW

*Salvia plebeia* R. Br. (SP) was collected and dried in June 2018 and purchased from Misan, Daegu, Korea. A reflux system of extraction was used to prepare water extract from SP. One liter of water was added to 50 g of dried SP. Additionally, the mixture was heated and extracted for 3 h at 250 °C. After the supernatant was obtained via centrifugation, it was filtered using a Whatman filter paper No. 6. The filtrate was frozen overnight, freeze-dried and stored at −20 °C until use. The extraction yield was 14.80 ± 1.90%.

### 2.2. Measurement of Total Phenolic Compounds and Flavonoid Content

The phenolic compound of SPW were measured by the Folin–Ciocalteu method of Meda et al. [19]. SPW was dissolved in deionized water. After diluting 1 mL of SPW solution 10 times with distilled deionized water (DDW), 1 mL of Folin–Ciocalteu reagent was dispensed and mixed well. After reacting for 5 min, 10 mL of 7% sodium carbonate (Na_2_CO_3_) and 4 mL of DDW were added to the mixture, and vortexed. After reacting at room temperature for 90 min, the absorbance of the mixture was measured at 750 nm using a fluorescence microplate reader (BioTek Instruments, Winooski, VT, USA). Gallic acid (Sigma–Aldrich, St. Louis, MO, USA) (0–300 µg/L) was used to produce the standard calibration curve. The value was calculated by mg gallic acid equivalents (GAE)/100 g SPW powder.

Flavonoid contents of SPW was determined by aluminum chloride (AlCl_3_) colorimetric assay [19,20]. After diluting 1 mL of SPW solution 5 times with DDW, dispense 0.3 mL of 5% sodium nitrate (NaNO_2_) and mix well. After reacting for 5 min, 0.3 mL of 5% aluminum chloride (AlCl_3_), 2 mL of 1 M sodium hydroxide (NaOH), and 2.4 mL DDW were added and vortexed. the absorbance of the mixture was measured at 510 nm using a fluorescence microplate reader. Catechin hydrate (Sigma-Aldrich, St. Louis, MO, USA) (0–300 µg/L) was used to produce the standard calibration curve. The value was calculated by mg catechin equivalents (CE)/100 g SPW powder.

### 2.3. Cell Culture and Cell Viability Assay

HepG2 cell line was purchased from the American Type Culture Collection (Manassas, VA, USA) to serve as the human liver carcinoma cell line. HepG2 cells were cultured in modified Eagle’s medium (MEM) supplemented with 10% fetal bovine serum and 1% penicillin/streptomycin in an environment of 5% carbon dioxide (CO_2_) at 37 °C. HepG2 cells were subcultured for 24 h in a 24-well cell culture plate (1 × 10^5^ cells/well), and at 70% confluence, they were pretreated with SPW for 24 h. Cytotoxicity was then measured using 2,3-bis [2-methoxy-4-nitro-5-sulfophen-yl]-2H-tetrazolium-5-carboxyanilide inner salt (XTT sodium salt) [21]. Briefly, 1 mg/mL XTT was dissolved in phenol red-free medium and mixed with 100 µM of peroxymonosulfate (PMS) in phosphate-buffered saline (PBS) (PMS:XTT, 1:800 ) to prepare the XTT working solution. After the pretreatment with SPW, the wells were washed with 1 mL PBS, followed by the addition of 1 mL PBS mixed with 250 µL of XTT working solution. After 2 h at 37 °C, the absorbance was measured at 450 nm using a microplate reader.

### 2.4. Oil Red O Staining

1 mM FFA solution was used to induce lipid accumulation in hepatocytes. Briefly, 1 mM FFA solution was generated by mixing oleic and palmitic acids in a ratio of 2:1 and then dissolving the mixture in MEM supplemented with 1% bovine serum albumin (BSA) to enable the formation of a complex between FFA and BSA [7]. This solution was treated for 4 h in HepG2 cells. Oil Red O reagent was dissolved in isopropanol overnight and then filtered. The filtrate was diluted 6:4 with distilled water (DW) overnight and filtered to prepare the working solution. Cells were fixed with 10% formalin for 30 min and then washed twice with DW. After lipid staining for 1 h using Oil Red O working solution, the cells were washed twice with DW. The stained cells were observed under a microscope (Leica DMi1; Leica Microsystems, Wetzlar, Germany), and the amount of stained cells was determined by measuring the absorbance at 510 nm by re-dissolving it in isopropanol.

### 2.5. AdipoRed Assay

To induce the triglyceride (TG) accumulation in hepatocytes, 1 mM FFA solution was used for 4 h. The cells were washed with 1 mL PBS. Next, 0.5 mL PBS was dispensed into each well. The cells were treated with AdipoRed^TM^ reagent (Lonza, Basel, Switzerland) for 15 min, and then the supernatant was removed. The degree of staining was measured by measuring the fluorescence using a microplate reader at an excitation wavelength of 485 nm and an emission wavelength of 530 nm.

### 2.6. Animals and Experimental Design

Male C57BL/6N mice (age, 7-week-old) were purchased from Orient Bio (Seongnam, Korea). Mice were acclimatized to ad libitum food and water intake in a 12 h light/dark cycle under constant humidity (40–60%) and temperature (20–24 °C). After 1 week of adaptation, the mice were randomly divided into 3 groups: (1) CON, mice fed a normal diet (AIN-76A diet; Research Diet, Inc., New Brunswick, NJ, USA); (2) HFD, mice fed 60% kcal HFD (D12492; Research Diet, Inc., New Brunswick, NJ, USA); and (3) HFD + SPW, mice fed 60% kcal HFD supplemented with SPW (200 mg/kg b.w./day). The animal feed of the HFD + SPW group was repelletized after mixing HFD and SPW. The experiment was conducted for 12 weeks, and the diet consumed during the experiment was measured daily. Body weight was also measured twice per week. After 12 weeks of feeding diet intake, animals were starved overnight and sacrificed under the anesthesia. All parts of the experiment were approved by the Institutional Animal Care and Use Committee (IACUC) of Chonnam National University (Approval No. CNU IACUC-YB-2021-85).

### 2.7. Oral Glucose Tolerance Test (OGTT)

At 11 weeks of the experiment, 1 week before sacrifice, mice in each group had fasted for 16 h and the blood glucose levels were measured the next day. For OGTT, 1.5 g/kg body weight of glucose was orally administered. Glucose levels were measured using a glucose monitor (CareSens N; i-sens, Inc., Seoul, Korea) in blood obtained from the tail vein every 30 min for 2 h, and the measured values were used to plot a graph. The area under the curves (AUC) was calculated from glucose levels over time.

### 2.8. Histological Analysis of the Liver

After the liver was collected from each mouse, it was immersed in 10% formalin and fixed. The sample was then converted into a paraffin block and cut at 10-µm intervals. After each compartment was stained with hematoxylin and eosin (H&E), the sections were observed under a microscope (Leica DMil; Leica Microsystems, Wetzlar, Germany).

### 2.9. Biochemical Assays

Blood from the postcaval vein of each mouse was collected in blood collection tubes and centrifuged at 3000 rpm for 15 min at 4 °C. The resulting serum was stored at −80 °C until use. Glutamic oxaloacetic transaminase (GOT), glutamic pyruvic transaminase (GPT), total cholesterol (TC), TG, and high-density lipoprotein cholesterol (HDL) assay kits were purchased from Asan Pharmaceutical (Seoul, Korea), and low-density lipoprotein cholesterol (LDL) value was calculated using Friedewald’s formula. Non-esterified fatty acid (NEFA) assay kit (NEFA-Wako; Wako Pure Chemical Industries, Osaka, Japan) was used to measure the level of serum fatty acid. The liver was quantified in 0.5 g and homogenized with 5 mL PBS using a glass-Teflon homogenizer (Daihan, Wonju, Korea). Hepatic lipids were extracted from the liver homogenate using the Folch method [22], and TC and TG assays were performed with commercial kits. 

### 2.10. Total RNA Isolation and Real-Time Polymerase Chain Reaction

The easy-BLUE^TM^ total RNA extraction kit (iNtRON, Seungnam, Korea)was used to extract total RNA from cells and the liver, and the amount of extracted RNA was quantified using nanodrop (JC bio, Seoul, Korea). Total RNA was reverse-transcribed to complementary DNA (cDNA) using the iScript^TM^ cDNA synthesis kit (Bio-Rad Laboratories, Hercules, CA, USA). Custom primers, iQ^TM^ SYBR^®^ Green Supermix (Bio-Rad Laboratories, Hercules, CA, USA), and cDNA were dispensed in a 96-well polymerase chain reaction (PCR) plate. Real time (RT)-PCR was conducted using the CFX96 Touch real-time PCR detection system (Bio-Rad Laboratories). The cDNA was subjected to 40 cycles of amplification, denaturation (95 °C for 30 s), annealing (58 °C for 30 s), and extension (72 °C for 45 s). The custom primer sequences are shown in Table 1.

### 2.11. Statistical Analysis

All data are expressed as mean ± standard deviation (SD) for the in vitro experiments and mean ± standard error (SE) for the in vivo experiments. One-way analysis of variance (ANOVA) was used to carry out statistical analysis, and the results were expressed based on a comparison to the result obtained for Duncan’s multiple range test (*p* < 0.05) or Student’s *t*-test using SPSS software (version 26.0, IBM Corporation, Armonk, NY, USA).

## 3. Results

### 3.1. Total Phenolic Compounds and Flavonoid Content of SPW was Determinated

Phenolic compounds (caffeic acid, rosmarinic acid, etc.) and flavonoid (luteolin, etc.), which are abundantly contained in SP, are known to be effective in NAFLD in prior studies [23,24,25]. When SP was extracted with SPW through reflux extraction system, the quantification of the phenolic compounds and flavonoids contents can be used as an index to predict how much activity this extract has that can alleviate NAFLD. Total phenolic compounds content was 6984.69 ± 138.25 mg GAE/100 g SPW and flavonoid content was 5334.44 ± 172.78 mg CE/100 g SPW (Table 2). Therefore, this study was tried to estimate the effect of SPW on NAFLD.

### 3.2. SPW Ameliorates FFA-Induced Lipid Accumulation

The purpose of this study was to investigate the mechanism whereby SPW alleviates hepatic steatosis using HepG2 cells, a human hepatocellular carcinoma cell line. Because it was confirmed that there was no cytotoxicity when HepG2 cells were treated with 1 mM FFA and then cultured for more than 20 h in various previous study [26,27,28], to induce a state of intracellular lipid accumulation, 1 mM FFA–BSA complex was administered to cells for 4 h, and the lipid accumulation-inhibitory effect of SPW was examined [29]. When the cytotoxicity of SPW was measured, no cytotoxicity was observed up to the concentration of 1000 µg/mL (Figure 1A). Intracellular TG level measured by AdipoRed^TM^ assay was found to significantly increase with FFA treatment, while a significant decrease was found with 100 µg/mL SPW (Figure 1B). When lipid accumulation was determined by staining all intracellular hydrophobic groups using Oil Red O stain, lipid accumulation was significantly increased in the FFA group and significantly decreased in the SPW 100 µg/mL treatment group (Figure 1C). The Oil Red O stained-section was observed under a microscope. The concentration of stained lipids increased with FFA treatment, while decreasing with SPW treatment (Figure 1D). These results indicated that SPW significantly inhibited lipid accumulation in the FFA-induced HepG2 cells.

### 3.3. SPW Regulates Lipid Metabolism via the AMPK Pathway in FFA-Induced Hepatic Steatosis

To investigate the inhibitory effect of SPW on intracellular lipid accumulation in the liver, the mRNA expression levels were analyzed in HepG2 cells treated with 1 mM FFA solution for 6 h. Lipid uptake markers, such as CD36, FATP2, and FATP5, were significantly increased in the FFA group compared to that in the CON group. On the other hand, CD36 significantly decreased with 100 µg/mL of SPW treatment, and FATP2 and FATP5 decreased from 50 µg/mL of SPW treatment (Figure 2A). The expression level of AMPK, which was decreased in the FFA group, was significantly increased in the SPW treatment groups (Figure 2B). When the expression levels of markers related to β-oxidation and synthesis were estimated, CPT-1 and PPARα were decreased in the FFA group, while SREBP-1c, FAS, and ACC were increased. In the SPW-treated group, markers of β-oxidation increased, and lipid synthesis factors decreased, confirming the mitigation of lipid accumulation (Figure 2C,D).

### 3.4. SPW Improves HFD-Induced Body Weight Gain

To effectively induce hepatic steatosis and obesity, mice were fed an HFD for 12 weeks. The weight gain of HFD-fed mice (HFD group) was significantly increased compared to that of normal diet-fed mice (CON group); however, compared to HFD-fed mice, SPW-fed mice (HFD + SPW group) displayed a significant decrease in weight gain. Food intake was significantly lower in the HFD and HFD + SPW groups than in the CON group; however, no significant difference was found based on the energy intake. Further, the food efficiency ratio was significantly higher and lower in the HFD and HFD + SPW groups than in the CON group, respectively. As body weight increased owing to HFD consumption, the weights of liver, peripheral, and epididymal fats significantly increased in the HFD group; however, these symptoms were relieved in the HFD + SPW group (Table 3). These results suggested that HFD-induced weight gain and liver weight gain can be alleviated by SPW intake.

### 3.5. SPW Improves HFD-Induced Glucose Level

NAFLD and insulin resistance are closely related. Insulin resistance is known to indicate marked visceral fat, which mainly accompanies obesity [30]. The HFD-induced increase in body weight was found to significantly increase fasting blood glucose level in the HFD group; however, this increase was significantly alleviated in the HFD + SPW group (Figure 3A). Further, when an OGTT was performed to measure insulin resistance, the ability of the HFD group to dispose of a glucose load was found to decrease, while this ability was enhanced in the HFD + SPW group (Figure 3B). Therefore, the results indicated that the abnormality of changes in blood glucose levels was closely related to NAFLD and HFD intake and could be alleviated by SPW intake.

### 3.6. SPW Ameliorates HFD-Induced Hepatic Steatosis and Liver Injury

The purpose of this study was to evaluate the morphological changes and accumulation of lipid droplets in the liver to determine whether SPW inhibited hepatic steatosis and liver injury. When the liver tissue was observed, its size and color changed in the HFD group compared to those in the CON group; these changes were mitigated in the HFD + SPW group (Figure 4A). Similarly, based on H&E staining of the liver, the number of lipid droplets was larger in the HFD group than that in the CON group, suggesting the occurrence of lipid accumulation. The number of lipid droplets in the HFD + SPW group was less than that in the HFD group (Figure 4B). Measurements of the biochemical parameters in the blood of the HFD group revealed that TG, TC, and LDL levels increased as NEFA levels increased, whereas HDL level decreased. Notably, the levels of NEFA, TG, TC, LDL were decreased in the HFD + SPW group and the level of HDL was increased (Table 4). Based upon these results, it was suggested that lipid accumulation was alleviated in the HFD + SPW group.

Aspartate aminotransferase (AST) and alanine transaminase (ALT) are enzymes in hepatocytes that leak into the bloodstream, and their blood levels are elevated in liver damage [31]. AST and ALT levels were higher in the HFD group than in the CON group, indicating liver damage. In the HFD + SPW group, the levels of AST and ALT significantly decreased, indicating the alleviation of liver damage by SPW (Figure 4C). Collectively, these findings suggested that HFD-induced intrahepatic steatosis and liver damage were significantly alleviated by SPW.

### 3.7. SPW Regulates Lipid Metabolism via AMPK Pathway in HFD-Induced Hepatic Steatosis

Measuring the expression levels of CD36, FATP2, and FATP5 related to fatty acid uptake revealed that compared to the CON group, the HFD group presented increased levels of these markers, while the HFD + SPW group exhibited significantly decreased levels (Figure 5C). AMPK, a factor regulating lipid metabolism, was significantly down-regulated in the HFD group and upregulated in the HFD + SPW group. By regulating mRNA expression, the protein expression level of AMPK was also decreased in the HFD group, and this decrease was alleviated in the HFD + SPW group. In addition, when AMPK phosphorylation was confirmed, it was significantly decreased in the HFD group compared to the CON group and significantly increased in the HFD + SPW group (Figure 5A,B). This result is not only an expression of AMPK but also phosphorylation of AMPK affected lipid metabolism. The expression levels of liver X receptor α (LXRα) and liver X receptor β (LXRβ), which regulate lipid synthesis, were increased in the HFD group and suppressed in the HFD + SPW group (Figure 5D). Levels of CPT-1 and PPARα, which are involved in β-oxidation, decreased in the HFD group and significantly increased in the HFD + SPW group (Figure 5E). Expression of SREBP-1c, FAS, and ACC, which are involved in lipid synthesis, increased in the HFD group and significantly decreased in the HFD + SPW group (Figure 5F). Overall, SPW intake might attenuate NAFLD by modulating AMPK-mediated lipid metabolism.

## 4. Discussion

The liver plays a central role in the metabolic processes of the body, including energy and nutrient metabolism, bile synthesis, bilirubin metabolism, blood coagulation, and detoxification of drugs and toxins [32,33]. Several nutrients are metabolized in the liver, including carbohydrates, lipids, and proteins. NAFLD is closely related to lipid metabolism in the liver. Under a normal metabolic state, the regulation of lipid metabolism occurs due to the action of blood glucose and insulin. If the nutrient excess state continues due to excessive intake of fat, the levels of FFAs in the blood increase [34,35]. As this condition continues, insulin resistance is induced, blood insulin sensitivity is reduced, and the hyperglycemic state continues [36]. In a hyperglycemic state, the decomposition of adipose tissue is accelerated, the concentration of FFAs in the blood is further increased, and the level of FFAs flowing into the liver are elevated [35]. Therefore, FFAs introduced into the liver induce the synthesis of TGs and inhibits β-oxidation, thereby increasing the level of hepatocellular lipid accumulation [37].

To proceed with this study, the HFD-induced mice model was determined as the NAFLD model. The rising risk of NAFLD is paralleled by lipid homeostasis that is disturbed by the occurrence of obesity [38]. Accordingly, various preclinical models related to obesity were researched, but this study used a model in which NAFLD was induced by ingesting an HFD [39,40]. In the current study, HFD induced weight gain, increase in ALT, AST, TG, TC, LDL, and decrease in HDL in the mice model. The increase in liver weight, intrahepatic TG, TC, and decrease in HFD mean that obesity and NAFLD were induced. SPW decreased intrahepatic TG and TC and increased HDL with weight loss. Therefore, this study was conducted to estimate the effect of SPW on the pathogenesis of NAFLD due to the intake of HFD and to identify the related pathway.

In a previous study, morphological observation using H&E staining of mouse liver and AdipoRed^TM^ assay and Oil Red O staining of HepG2 cells revealed lipid accumulation in the liver [41]. Based upon H&E staining, larger and more lipid droplets were observed in the HFD group; however, these were decreased in the HFD + SPW group. Additionally, SPW treatment inhibited intracellular TGs and lipid accumulation based on a reduction of the AdipoRed^TM^ and Oil Red O staining area in HepG2 cells. These results indicated that SPW was confirmed to effectively reduce lipid accumulation in mice and HepG2 cells.

Hepatic steatosis can be determined through changes in body weight and biochemical parameters [42]. In our study, when hepatic steatosis was induced via HFD, body, liver, and fat weights were found to increase. However, they were significantly decreased in the HFD + SPW group. Additionally, the levels of TG, TC, LDL, and HDL were improved by SPW. In the hepatic steatosis conditions, the ability to decompose a glucose load decreases, which in turn, lipid metabolism disorder and insulin resistance occurs [43]. Our results showed that fasting blood glucose level was decreased in HFD + SPW mice. These suggested that SPW intake alleviated the insulin resistance caused by this abnormal lipid metabolism and controlled the concentration of FFAs to suppress fat accumulation in the liver.

The influx of FFAs induces lipid accumulation in the liver. This influx into the liver is regulated by the lipid transporters CD36, FATP2, and FATP5. The lipid transporter, CD36, plays a role in intracellular fatty acid uptake signaling. Furthermore, the fatty acid transport protein family members, FATP2 and FATP5, have an extracellular binding site, acyl-CoA synthetase active site, and adenosine triphosphate (ATP) binding domain [44,45,46]. In a previous study, when FATP2 and FATP5 were specifically knocked down in the liver, the absorption of long-chain fatty liver was reduced, the formation of TG and lipid droplets was suppressed, and hepatic steatosis and insulin resistance were improved [47]. In this study, the expression of CD36, FATP2, and FATP5 related to lipid uptake increased and lipid accumulation intensified. However, hepatic steatosis was suggested to be alleviated by SPW in the HFD-fed mice model and the FFA-treated cell model.

The alleviation of hepatic steatosis by SPW supplementation could be caused by the regulation of AMPK-induced lipid synthesis and β-oxidation. AMPK plays an important role in maintaining intracellular energy homeostasis and is known to be closely related to hepatic steatosis [48]. In previous studies, the phosphorylation of AMPK affected the phosphorylation and inactivation of ACC, and depletion and accumulation of FAS in the cytoplasm [8,49]. The dephosphorylation and activation of ACC promote the production of malonyl-CoA, which can suppress the expression of CPT-1, which regulates β-oxidation [50]. In addition, AMPK has been confirmed to phosphorylate SREBP-1c to induce the inhibition of proteolytic processing and transcriptional activity, thereby reducing the expression of lipid synthase. In addition, LXR has subtypes (α and β) and is a nuclear receptor that enhances the transcription of genes that regulate fatty acid synthesis and cholesterol efflux [51,52]. In particular, LXRα has been found to be specifically present in metabolically active tissues, such as the liver and adipose tissue, and its activity is inhibited as the threonine residue of AMPK is phosphorylated [10]. In a previous study, the LXR agonists were controlled hypertriglyceridemia or hepatic steatosis by increasing the expression of SREBP-1c [53]. Our recent results revealed that AMPK and LXR are regulated by SPW; thus, CPT-1 and PPARα, which regulate the β-oxidation of lipids, were significantly elevated; and SREBP-1c, FAS, and ACC, which regulate lipid synthesis, were antagonized.

## 5. Conclusions

In the present study, we demonstrated that SPW could effectively alleviate NAFLD by inhibiting lipid accumulation. In the signaling pathway related to lipid accumulation, CD36, FATP2, and FATP5, which were involved in lipid uptake, were decreased, thereby reducing the influx of FFA. SPW inhibited lipid accumulation modulating the level of AMPK, FXR, and LXR which regulated lipid metabolism decreasing the level of SREBP1-c, ACC, and FAS, which were related to fatty acid synthesis, and increasing the level of CPT-1 and PPARα which were related to β-oxidation (Figure 6). Based on results of this study, SPW could be a useful NAFLD-alleviating functional material.

## Figures and Tables

**Figure 1 nutrients-14-05379-f001:**
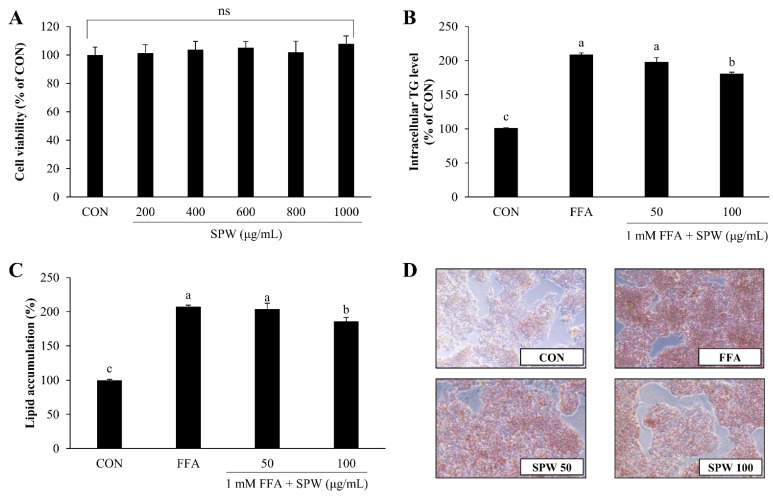
*Salvia plebeia* R. Br. water extract (SPW) treatment decreased lipid accumulation in free fatty acid-induced HepG2 cells. (**A**) Cytotoxicity was determined using the XTT assay. (**B**) Intracellular triglyceride was measured using the AdipoRed^TM^ assay, and (**C**) lipid accumulation was measured using Oil Red O staining. Quantitative lipid accumulation of Oil Red O contents at 500 nm. (**D**) Oil Red O staining image of HepG2 cells. Data are expressed as mean ± SD (*n* = 4 for each group). CON, control; FFA, free fatty acid; TG, triglyceride; XTT, 2,3-bis [2-methoxy-4-nitro-5-sulfophen-yl]-2H-tetrazolium-5-carboxyanilide; SD, standard deviation; ns., not significant; SPW 50, FFA + SPW 50 µg/mL; SPW 100, FFA + SPW 100 µg/mL. Different letters above the bar indicate statistical difference based on the Duncan’s multiple range test (*p* < 0.05, a > b > c).

**Figure 2 nutrients-14-05379-f002:**
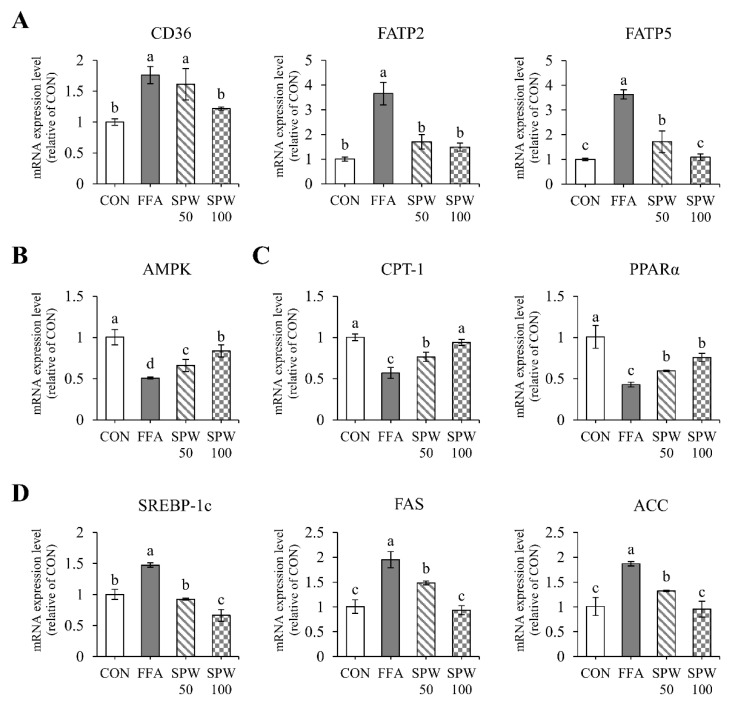
*Salvia plebeia* R. Br. water extract (SPW) treatment regulated the expression of genes related to lipid metabolism in FFA-induced HepG2 cells. The mRNA expression level of (**A**) lipid uptake markers, CD36, FATP2, and FATP5; (**B**) lipid metabolism markers, AMPK, LXRα, and LXRβ; (**C**) β-oxidation markers, CPT-1 and PPARα; and (**D**) lipid synthesis markers, SREBP-1c, FAS, and ACC. Data are expressed as mean ± SD (*n* = 4 for each group). CON, control; FFA, free fatty acid; SPW 50, FFA + SPW 50 µg/mL; SPW 100, FFA + SPW 100 µg/mL; CD36, cluster of differentiation 36; FATP2, fatty acid transport protein 2; FATP5, fatty acid transport protein 5; AMPK, AMP-activated protein kinase; LXRα, liver X receptor α; LXRβ, liver X receptor β; CPT-1, carnitine palmitoyl transferase-1α; PPARα, peroxisome proliferator-activated receptors α; SREBP-1c, sterol regulatory element binding protein-1c; FAS, fatty acid synthase; ACC, acetyl-CoA carboxylase. Different letters above the bar indicate statistical difference based on the Duncan’s multiple range test (*p* < 0.05, a > b > c > d).

**Figure 3 nutrients-14-05379-f003:**
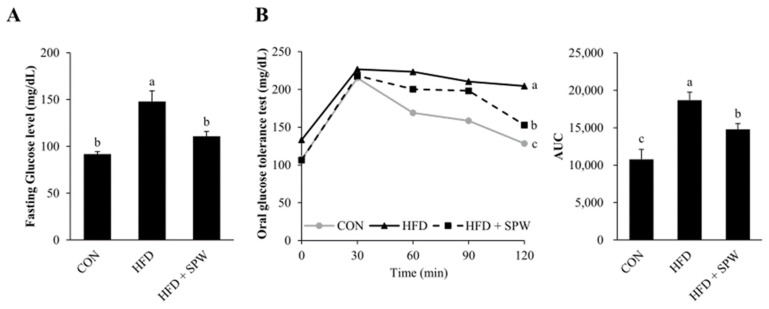
*Salvia plebeia* R. Br. water extract (SPW) intake improved glucose levels in HFD-induced NAFLD mice. After mice were subjected to a 16 h fasting period before the experiment, (**A**) fasting glucose levels and (**B**) glucose levels were measured for a total of 2 h at 30 min intervals in for OGTT. Data represent mean ± SE (*n* = 5 for each group). AUC, area under the curve; CON, control; HFD, high-fat diet; HFD + SPW, HFD + 200 mg/kg/day SPW; OGTT, Oral Glucose Tolerance Test; SE, standard error. The different letters within a column indicate statistical difference based on the Duncan multiple range test (*p* < 0.05, a > b > c).

**Figure 4 nutrients-14-05379-f004:**
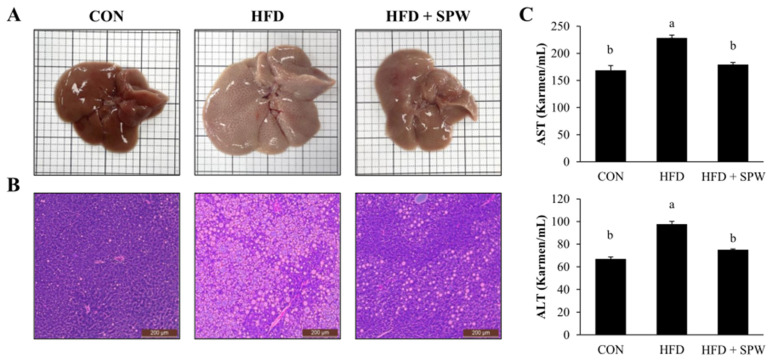
*Salvia plebeia* R. Br. water extract (SPW) intake ameliorated liver injury in HFD-induced NAFLD mice. (**A**) H&E stained liver tissues and (**B**) liver histology. (**C**) AST and ALT levels were measured in mice. Data represent mean ± SE (*n* = 5 for each group). AST, aspartate transaminase; ALT, alanine transaminase; CON, control; HFD, high-fat diet; HFD + SPW, HFD + 200 mg/kg/day SPW; H&E, hematoxylin and eosin. The different letters above the bar indicate statistical difference based on Duncan’s multiple range test (*p* < 0.05, a > b).

**Figure 5 nutrients-14-05379-f005:**
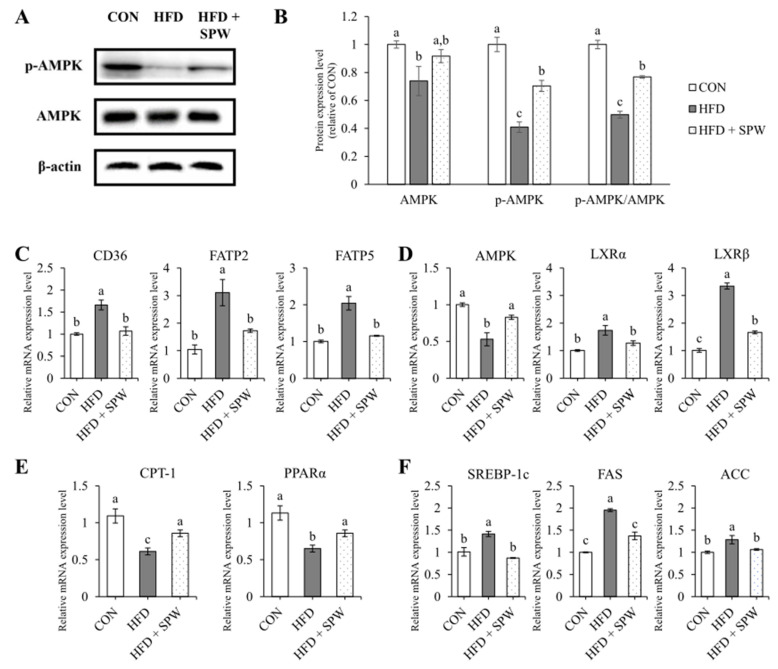
*Salvia plebeia* R. Br. water extract (SPW) intake regulated the gene expression of lipid metabolism factors in HFD-induced NAFLD mice. The protein expression level was measured by Western blotting. β-actin was used as standard for normalizing the expression. (**A**) Representative Western blot of AMPK, p-AMPK, and β-actin. (**B**) Normalized protein expression level. The mRNA expression level of (**C**) lipid uptake markers, CD36, FATP2, and FATP5; (**D**) lipid metabolism marker, AMPK; (**E**) lipid synthesis markers, SREBP-1c, FAS, and ACC; and (**F**) β-oxidation markers, CPT-1 and PPARα. Data represents the mean ± SE (standard error, *n* = 5 for each group): CON, control; HFD, high-fat diet; HFD + SPW, HFD + 200 mg/kg/day SPW; NAFLD, non-alcoholic fatty liver disease; p-AMPK; phospho-AMPK. The different letters above the bar indicate statistical difference based on the Duncan multiple range test (*p* < 0.05, a > b > c).

**Figure 6 nutrients-14-05379-f006:**
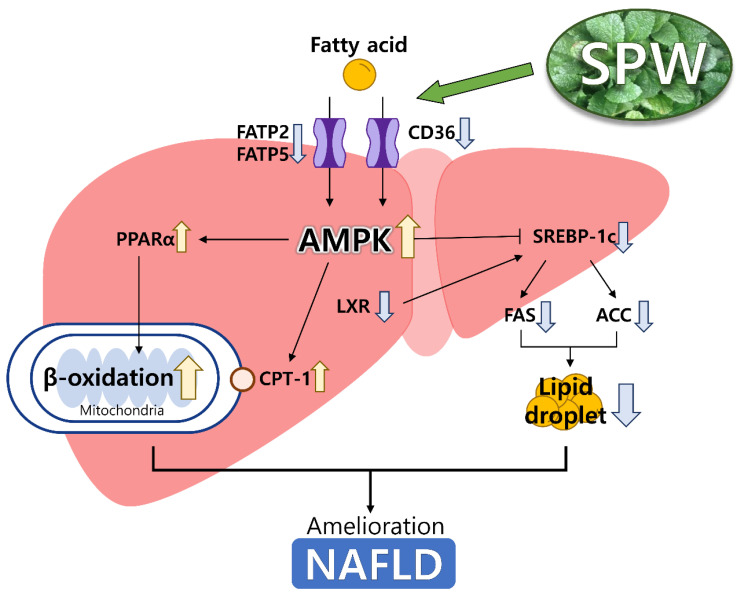
Scheme for NAFLD pathway in the liver. Yellow arrows were meant up-regulated the expression and blue arrows were meant down-regulated the expression.

**Table 1 nutrients-14-05379-t001:** Sequence of the primers used for real-time PCR.

Gene		Primer Sequence (5′ to 3′)
*hCD36*	Forward Reverse	5′-TGGAACAGAGGCTGACAACT-3′ 5′-TTGATTTTTGATAGATATGGG-3′
*hFATP2*	Forward Reverse	5′-CTTTCAGCACATTGCTGATTACCT-3′ 5′-CAGTGATCTCAATGGTGTCCTGTAT-3′
*hFATP5*	Forward Reverse	5′-AGCTCCTGCGGTACTTGTGT-3′ 5′-AAGGTCTCCCACACATCAGC-3′
*hAMPK*	Forward Reverse	5′-GGCACCCTCCCATTTGATG-3′ 5′-ACACCCCCTCGGATCTTCTT-3′
*hAMPK*	Forward Reverse	5′-GGCACCCTCCCATTTGATG-3′ 5′-ACACCCCCTCGGATCTTCTT-3′
*hCPT-1*	Forward Reverse	5′-TGTTGGGTATGCTGTTCATGACA-3′ 5′-GCGGCCTGGGTAGGAAGA-3′
*hPPARα*	Forward Reverse	5′-AACATCCAAGAGATTTCGCAATC-3′ 5′-CCGTAAAGCCAAAGCTTCCA-3′
*hSREBP-1c*	Forward Reverse	5′-CGGAACCATCTTGGCAACA-3′ 5′-GCCGGTTGATAGGCAGCTT-3′
*hFAS*	Forward Reverse	5′-CGCTCGGCATGGCTATCT-3′ 5′-CTCGTTGAAGAACGCATCCA-3′
*hACC*	Forward Reverse	5′-TGCAGATCTTAGCGGACCAA-3′ 5′-GCCTGCGTTGTACAGAGCAA-3′
*hβ-Actin*	Forward Reverse	5′-ACGGCCAGGTCATCACTATTG-3′ 5′-CAAGAAGGAAGGCTGGAAAAGA-3′
*mCD36*	Forward Reverse	5′-TTGAAGGCATTCCCACGTATC-3′ 5′-CGGACCCGTTGGCAAA-3′
*mFATP2*	Forward Reverse	5′-GGAACCACAGGTCTTCCAAA-3′ 5′-TAAAGTAGCCCCAACCACGA-3′
*mFATP5*	Forward Reverse	5′-GGAACCACAGGTCTTCCAAA-3′ 5′-TAAAGTAGCCCCAACCACGA-3′
*mAMPK*	Forward Reverse	5′-TTCGTGCCGCCCCTTT-3′ 5′-GGTCAGCATGCCCACAAAA-3′
*mLXRα*	Forward Reverse	5′-CCTCTGGCTTCCATTACAAC-3′ 5′-CTTCTGACAGCACACACTC-3′
*mLXRβ*	Forward Reverse	5′-CACCATTGAGATCATGTTGC-3′ 5′-TTGATCCTCGTGTAGGAGAG-3′
*mCPT-1*	Forward Reverse	5′-GTGACTGGTGGGAGGAATAC-3′ 5′-GAGCATCTCCATGGCGTAG-3′
*mPPARα*	Forward Reverse	5′-TGGCAAAAGGCAAGGAGAAG-3′ 5′-CCCTCTACATAGAACTGCAA-3′
*mSREBP-1c*	Forward Reverse	5′-TGGCTTGGTGATGCTATGTTG-3′ 5′-GACCATCAAGGCCCCTCAA-3′
*mFAS*	Forward Reverse	5′-GAAGTGTCTGGACTGTGTCATTTTTAC-3′ 5′-TTAATTGTGGGATCAGGAGAGCAT-3′
*mACC*	Forward Reverse	5′-TCCCCAAGTTCTTCACGTTCA-3′ 5′-CAGGCTCCAAGTGGCGATAA-3′
*mβ-Actin*	Forward Reverse	5′-AGCCATGTACGTAGCCATCC-3′ 5′-CTCTCAGCTGTGGTGGTGAA-3′

PCR, polymerase chain reaction.

**Table 2 nutrients-14-05379-t002:** Total phenolic compounds and flavonoid content of *Salvia plebeia* R. Br. water extract (SPW).

	Total Phenolic Compounds Content (mg GAE/100 g SPW)	Flavonoid Content (mg CE/100 g SPW)
**SPW**	6984.69 ± 138.25	5334.44 ± 172.78

SPW, *Salvia plebeia* R. Br. water extract; GAE, gallic acid equivalent; CE, catechin equivalent.

**Table 3 nutrients-14-05379-t003:** Regulatory effect of SPW on simple steatosis in high-fat diet-induced NAFLD mice.

Groups	CON	HFD	HFD + SPW
Body mass (g)			
Initial (A)	22.78 ± 0.37 ^ns^	22.79 ± 0.33	22.76 ± 0.28
Final (B)	35.45 ± 0.66 ^c^	45.81 ± 0.67 ^a^	43.36 ± 0.61 ^b^
Weight gain (B–A)	12.67 ± 0.72 ^c^	23.10 ± 0.55 ^a^	20.61 ± 0.58 ^b^
Food intake (g/day)	3.23 ± 0.03 ^a^	2.52 ± 0.04 ^b^	2.66 ± 0.05 ^b^
Energy intake (kcal/day)	12.60 ± 0.13 ^ns^	13.19 ± 0.19	13.93 ± 0.25
FER (%)	4.62 ± 0.27 ^c^	10.61 ± 0.26 ^a^	9.19 ± 0.31 ^b^
Liver weight (g)	1.44 ± 0.02 ^b^	1.94 ± 0.05 ^a^	1.55 ± 0.12 ^b^
Liver index (%)	3.68 ± 0.04 ^b^	4.19 ± 0.10 ^a^	3.09 ± 0.21 ^c^
Relative fat weight (g)			
Perirenal fat	0.75 ± 0.03 ^c^	1.39 ± 0.07 ^a^	1.08 ± 0.06 ^b^
Epididymal fat	1.49 ± 0.03 ^b^	2.12 ± 0.12 ^a^	1.62 ± 0.05 ^b^

Data represent mean ± SE (*n* = 5 for each group). FER, food efficiency ratio; ns, not significant; CON, control; HFD, high-fat diet; HFD + SPW, HFD + 200 mg/kg/day *Salvia plebeia* R. Br. water extract (SPW); NAFLD, non-alcoholic fatty liver disease; SE, standard error. The different letters within a column indicate statistical difference based on the Duncan multiple range test (*p* < 0.05, a > b > c).

**Table 4 nutrients-14-05379-t004:** Biochemical parameters in HFD-induced NAFLD mice.

Groups	CON	HFD	HFD + SPW
Serum			
TG (mg/dL)	101.79 ± 1.05 ^c^	134.23 ± 3.51 ^a^	118.16 ± 1.91 ^b^
TC (mg/dL)	239.44 ± 1.27 ^b^	278.38 ± 5.56 ^a^	237.44 ± 3.22 ^b^
HDL (mg/dL)	122.12 ± 1.72 ^b^	95.75 ± 0.72 ^a^	119.55 ± 2.40 ^b^
LDL (mg/dL)	96.96 ± 2.06 ^b^	155.78 ± 5.13 ^a^	96.25 ± 4.79 ^b^
NEFA (mEq/L)	80.33 ± 2.66 ^b^	120.33 ± 5.61 ^a^	81.00 ± 3.22 ^b^
Liver tissue			
TG (mg/g liver)	5.77 ± 0.35 ^c^	10.73 ± 0.07 ^a^	8.93 ± 0.05 ^b^
TC (mg/g liver)	2.80 ± 0.06 ^b^	4.16 ± 0.12 ^a^	2.75 ± 0.09 ^b^

Data represent mean ± SE (*n* = 5 for each group). TG, Triglyceride; TC, Total cholesterol; HDL, High-density lipoprotein cholesterol; LDL, Low-density lipoprotein cholesterol; NEFA, Non-esterified fatty acid; CON, control; HFD, high-fat diet; HFD + SPW, HFD + 200 mg/kg/day SPW. The different letters within a column indicate statistical difference based on the Duncan multiple range test (*p* < 0.05, a > b > c).

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
