# Peer review of "Salvia plebeia R. Br. Water Extract Ameliorates Hepatic Steatosis in a Non-Alcoholic Fatty Liver Disease Model by Regulating the AMPK Pathway"

_nutrients, 2022, doi:10.3390/nu14245379_

Round 1
Reviewer 1 Report
This manuscript studied the effects of water extract of a plant, Salvia plebeian, on fatty liver disease by using two different experimental models, hepatocytes and mice. Topics is potentially significant, however, there are two major concerns for the manuscript.
First is the unknown composition of extract used. Although authors tried to provide information, it is major drawback of the study. It would be suggested to at least quantify unknown compounds from the extract, which would be beneficial for future studies to compare with the current extract used.
The other major concern for the study is authors primarily checked the transcription levels for all genes of interest. However, as authors clearly indicated in lines 353-358, key genes, including AMPK, PPAR-alpha, SREPB-1c, are well known to be regulated post-translationally, rather than transcription levels. Without knowing how these proteins respond to the treatment, it is not conclusive if this treatment elicited observed effects via AMPK specific mechanism as concluded by authors.
Information of diet used should be provided.
Justification of doses used for treatment needs to be included as well as how doses (200 mg/kg bw/day) were delivered.
Line 70, as this plant is used in other countries, it is suggested to be used in Asian countries.
Plant collection year should be included. It is not clear how water extract can be done at 250C. Also include yield of extract.
Line 117, please include information of company for AdipoRed.
Experimental protocol needs more detail information, such as when OGTT were performed (after feeding how many weeks), sacrifice conditions (fasted or not), which samples were used to determine blood glucose shown in Fig. 3A, how long cells were treated for results in Fig. 1B-D & 2, etc.
Lines 177-179, please clarify this sentence what observations you are describing.
Fig. 1A, there was no cell viability test of FFA only and 50 and 100 ug/ml SPW treatment. It is known that free fatty acid treatment itself would have cytotoxic effects in this cell line. Thus, it is necessary to check cell viability of FFA and/or SPW as used for experimental conditions.
Results in Table 2 shows that HFD+SPW treated animals had higher body weights than control (43g vs. 35g in control, ~8g difference) although less than HFD only group. However, HFD+SPW treated animals had similar size of liver and slightly heavier adipose tissue weights (~0.3 g difference from the control). It is not clear how HFD+SPW treated animals had such an increase in weights without corresponding increase of liver and adipose tissue. This needs to be discussed.
There was no information of Fig. 1D in legend.
It is not clear why authors used student t-test for only Food intake results in Table 2. Since there were 3 groups, t-test should not be used for any analysis.
Table 3, it should be ‘the different letters within a row…’. Also results from liver tissue should be mg TG or TC per weight of liver, not dL.
Author Response
Nov 29, 2022
Dear Editor-in-Chief:
The reviewers were thorough and helpful in their review of manuscript No. nutrients-2053837 concerning improvement effect of Salvia plebeia R. Br. aqueous extract against the non-alcoholic fatty liver disease model. We revised manuscripts which we believe have been strengthened after considering their comments.
Reviewer 1
The changes made as reviewer suggested are as follows;
- It would be suggested to at least quantify unknown compounds from the extract, which would be beneficial for future studies to compare with the current extract used. : We described that phenolic compound and flavonoid contents of the extract from salvia plebeia R. Br. were measured and added to the manuscript.
- key genes, including AMPK, PPAR-alpha, SREPB-1c, are well known to be regulated post-translationally, rather than transcription levels. : As reviewer suggested, we appended the measuring the protein expression level is the surest way to identify the disease alleviation mechanism. But, since mRNA is unconditionally preceded in the process of protein expression, it is considered that confirming the mRNA expression level will have a meaningful result. However, as described in the discussion, the mechanism was predicted to be regulated through the phosphorylation of AMPK, so the phosphorylation level of AMPK was additionally confirmed at the cytoplasm. The result of protein expression was added to figure 5.
- Justification of doses used for treatment needs to be included as well as how doses (200 mg/kg bw/day) were delivered. : The animal feed of the HFD+SPW group was repelletized after mixing HFD and SPW. Mice were fed ad libitum.
- How long cells were treated for results. : The treat time was recorded for each method, and in the RT-PCR data of the in vitro experiment, it was written in the first paragraph of results 3.2.
- cell viability of FFA and/or SPW as used for experimental conditions.: In various previous studies, it was confirmed that there was no cytotoxicity when HepG2 cells were treated with 1 mM FFA and then cultured for more than 20 h [1-3]. Also, when observing the image of Oil Red O staining in this study, no decrease in the number of cells was observed in the group treated with 1 mM of FFA and SPW, so it can be predicted that the cell viability was stable.
- It is not clear how HFD+SPW treated animals had such an increase in weights without corresponding increase of liver and adipose tissue. This needs to be discussed. : Obesity means a state in which a large amount of fat is accumulated in the body. In addition to perirenal fat and epididymal fat collected in this experiment, adipose tissue accumulation may occur in various body parts such as mesenteric and inguinal fat. In addition, lipid droplets observed in the liver, muscle, etc. are fat storage organelles in the cytoplasm that exist in most eukaryotic cells and can be easily observed in a wide range [4]. Therefore, it is not possible to identify the mechanism of weight loss only with the liver weight measured in this experiment and the weight of some adipose tissues. Also, when expressed as a weight-to-weight ratio, unlike testicular fat and liver index, perirenal fat was 3.03 ± 0.49% and 2.49 ± 0.40% in the HFD and HFD+SPW groups, respectively, and there was no significant difference. These results predict that SPW effectively mitigates hepatic steatosis and decreases body weight, and can effectively reduce lipid accumulation in some organs such as epididymal fat. However, as all adipose tissue in the body was not estimated, it is difficult for this study to be used to confirm the anti-obesity effect.
- Unit of hepatic TG or TC : It was entered incorrectly by mistake and was corrected after rechecking the data.
[1] Han L.P.; Sun B.; Li C.J.; Xie Y.; Chen L.M. Effect of celastrol on toll‑like receptor 4‑mediated inflammatory response in free fatty acid‑induced HepG2 cells. International Journal of Molecular Medicine 2018.
[2] Chen Y.-C.; Chen H.-J.; Huang B.-M.; Chen Y.-C.; Chang C.-F. Polyphenol-Rich Extracts from Toona sinensis Bark and Fruit Ameliorate Free Fatty Acid-Induced Lipogenesis through AMPK and LC3 Pathways. Journal of Clinical Medicine 2019, 8, 1664.
[3] Im A.-R.; Kim Y.H.; Lee H.W.; Song K.H. Water extract of Dolichos lablab attenuates hepatic lipid accumulation in a cellular nonalcoholic fatty liver disease model. Journal of medicinal food 2016, 19, 495-503.
[4] Jarc Jovičić E.; Petan T. Lipid Droplets and the Management of Cellular Stress. The Yale journal of biology and medicine 2019, 92, 435-452.
Overall, we thank the reviewers for their positive comments as well as constructive criticism on our manuscript. We believe the manuscript is now acceptable for publication.
Thank you.

Reviewer 2 Report
Authors Investigated function of Salvia plebeia R. Br. Water Extract on non-alcoholic fatty liver disease. Preliminary evidence has been obtained that the hydrolysate will affect the formation of fatty liver by AMPK pathway. However, the experimental design needs to be improved.
In every experiment, CON+SWP are needed,especially for mouse experiments. The authors must verify that the hydrolysate does not affect the liver of healthy rats fed a long-term normal diet.
Author Response
Nov 29, 2022
Dear Editor-in-Chief:
The reviewers were thorough and helpful in their review of manuscript No. nutrients-2053837 concerning improvement effect of Salvia plebeia R. Br. aqueous extract against the non-alcoholic fatty liver disease model. We revised manuscripts which we believe have been strengthened after considering their comments.
Reviewer 2
The changes made as reviewer suggested are as follows;
- In every experiment, CON+SWP are needed,especially for mouse experiments. : This study was conducted to estimate whether SPW can significantly alleviate the lipid accumulation in the hepatocyte by FFA and HFD. Therefore, the effect of attenuating hepatic steatosis in the SPW-only group in vitro experiment and the CON+SPW group in vivo experiment were not confirmed. However, cytotoxicity was not found when HepG2 was treated with 0-1000 ug/mL of SPW for 24 h in cell experiments. In addition, in animal experiments, when looking at the measurement results of AST and ALT, which are enzymes present in hepatocytes and whose levels increase as they leak into the blood during liver damage [5], the values that increased significantly by the intake of HFD decreased by the intake of SPW. Therefore, SPW is unlikely to be toxic when ingested. In addition, when Korean red ginseng and Salvia plebeia R. Br 30% ethanolic extracts were mixed at a ratio of 1:3 and fed to SD rats at 500, 1000, and 2000 mg/kg/day for 4 weeks, significant liver damage was not found in liver function tests [6]. As a result, it is drawback in my study that the liver toxicity was not directly confirmed when SPW was consumed for a long time in this study, but when the results of the previous study and this study are combined, it is considered that the toxicity of SPW has little effect on the mouse model.
[5] Giannini E.; Botta F.; Fasoli A.; Ceppa P.; Risso D.; Lantieri P.B.; Celle G.; Testa R. Progressive liver functional impairment is associated with an increase in AST/ALT ratio. Digestive diseases and sciences 1999, 44, 1249-1253.
[6] Seo H.W.; Suh J.H.; Kyung J.-S.; Jang K.H.; So S.-H. Subacute Oral Toxicity and Bacterial Mutagenicity Study of a Mixture of Korean Red Ginseng (Panax ginseng C.A. Meyer) and Salvia plebeia R. Br. Extracts. Toxicological Research 2019, 35, 215-224.
Overall, we thank the reviewers for their positive comments as well as constructive criticism on our manuscript. We believe the manuscript is now acceptable for publication.
Thank you.

Round 2
Reviewer 1 Report
Most of comments were properly addressed.
Few minor comments
Extraction : Since authors used water to extract the sample, extraction temperature should not have reached 250C as authors described, likely a little over 100C as this is the boiling point of water.
Please define GAE and CE before using these acronyms in section 3.1 and Table 2.
Author Response
Division of Food and Nutrition
Chonnam National University
77, Yongbong-ro, Buk-gu
Gwangju, 61186 Korea
Phone: +82-62-530-1337
E-mail: wjjun@chonnam.ac.kr
Dec 7, 2022
Dear Editor-in-Chief:
The reviewers were thorough and helpful in their review of manuscript No. nutrients-2053837 concerning improvement effect of Salvia plebeia R. Br. aqueous extract against the non-alcoholic fatty liver disease model. We revised manuscripts which we believe have been strengthened after considering their comments.
Reviewer 1
The changes made as reviewer suggested are as follows;
- Extraction: Since authors used water to extract the sample, extraction temperature should not have reached 250C as authors described, likely a little over 100C as this is the boiling point of water.
- SP went through a steaming process with steam at 120 to 250 degrees. That destroys or softens the cell wall to improve the extraction efficiency of active ingredients within the cell. So, it has efficiently separated the active ingredients of SP (phenolic compound and flavonoid, etc.) [1].
- Please define GAE and CE before using these acronyms in section 3.1 and Table 2.
- It was added in manuscript line 101-102, 109-110, and 211.
Reviewer 2
- In every experiment, CON+SWP are needed. It is not only about cytotoxicity. Other potential impacts need to be excluded, otherwise the author's conclusion cannot be supported.
- Before anything else, I apologize for not being able to fully understand your first revision and not be able to give you a proper answer. The main significance of this manuscript is to investigate the effect whether the functional components of Salvia plebeia Br. (SP) directly alleviate the pathogenesis of diseases. Various models are used to investigate the effect of NAFLD, but the HFD-induced mouse model is used as a model that can confirm the onset of steatosis and NASH in particular [2, 3]. Related content has been added to the journal's discussion (Lines 373-382). Therefore, in this experiment, the in vivo experimental group consumed HFD to induce the onset of NAFLD, and in the group of HFD+SPW, it was confirmed what effect it has on the disorder of lipogenesis and b-oxidation in the liver, which are the main causes of NAFLD. In addition, since the effect of SPW relieves hepatic steatosis by regulating hepatic lipogenesis and lipid β-oxidation, the experiment had to be conducted under conditions in which intrahepatic lipid accumulation could occur with HFD rather than normal eating habits.
SP contains various physiologically active compounds. Among them, since caffeic acid, rosmarinic acid, and luteolin are known to have an alleviating effect on NAFLD, it can be assumed that SPW also has an alleviating effect on NAFLD. Among them, in an article confirming the effect of Caffeic acid (CA), hepatic lipogenesis and inflammation were alleviated in the state of NAFLD induced by HFD, but no significant effect occurred when normal diet (ND) and CA were co-ingested. could [4]. In addition, in an article confirming the NAFLD-alleviating effect of ginkgo biloba extract 50 (GBE 50), in the group inducing NAFLD through HFD, the intake of GBE 50 had an effect of relieving insulin resistance and hepatic steatosis, but when ingested with ND, ND intake No significant difference with group occurred [5]. Rhodomyrtus tomentosa Hassk fruit phenolic rich extract (RTE) reduced hepatic steatosis and inflammation when ingested with HFD and showed significant differences in metabolite clustering related to hepatic metabolism. On the other hand, the ND group and the group receiving both ND and RTE did not show obvious metabolic differentiation [6]. In addition, according to the study by chunchun Ding, et al. and the study by Layanne C. C. Araujo, et al., when ND and extract were consumed simultaneously, the effects of glucose metabolism, insulin resistance, and inflammation were lower than when HFD and extract were simultaneously consumed [7, 8]. The results of these preliminary studies suggest that confirming the effect of extract when NAFLD is induced by HFD intake can effectively confirm the function of NAFLD. In various studies, simultaneous intake of HFD and extract is adopted to identify the effect of NAFLD [9-18], and the results of this study claim that SPW can identify that it has a mitigating effect on NAFLD.
- Gong L.; Huang L.; Zhang Y. Effect of steam explosion treatment on barley bran phenolic compounds and antioxidant capacity. Journal of agricultural and food chemistry 2012, 60, 7177-7184.
- Castro R.E.; Diehl A.M. Towards a definite mouse model of NAFLD. Journal of hepatology 2018, 69, 272-274.
- Nakamura A.; Terauchi Y. Lessons from mouse models of high-fat diet-induced NAFLD. International journal of molecular sciences 2013, 14, 21240-21257.
- Mu H.-N.; Zhou Q.; Yang R.-Y.; Tang W.-Q.; Li H.-X.; Wang S.-M.; Li J.; Chen W.-X.; Dong J. Caffeic acid prevents non-alcoholic fatty liver disease induced by a high-fat diet through gut microbiota modulation in mice. Food Research International 2021, 143, 110240.
- Li L.; Yang L.; Yang F.; Zhao X.-l.; Xue S.; Gong F.-h. Ginkgo biloba extract 50 (GBE50) ameliorates insulin resistance, hepatic steatosis and liver injury in high fat diet-fed mice. Journal of Inflammation Research 2021, 14, 1959.
- Wang R.; Yao L.; Lin X.; Hu X.; Wang L. Exploring the potential mechanism of Rhodomyrtus tomentosa (Ait.) Hassk fruit phenolic rich extract on ameliorating nonalcoholic fatty liver disease by integration of transcriptomics and metabolomics profiling. Food Research International 2022, 151, 110824.
- Ding C.; Zhao Y.; Shi X.; Zhang N.; Zu G.; Li Z.; Zhou J.; Gao D.; Lv L.; Tian X. New insights into salvianolic acid A action: Regulation of the TXNIP/NLRP3 and TXNIP/ChREBP pathways ameliorates HFD-induced NAFLD in rats. Scientific reports 2016, 6, 1-12.
- Araujo L.C.; Feitosa K.B.; Murata G.M.; Furigo I.C.; Teixeira S.A.; Lucena C.F.; Ribeiro L.M.; Muscará M.N.; Costa S.K.; Donato J. Uncaria tomentosa improves insulin sensitivity and inflammation in experimental NAFLD. Scientific reports 2018, 8, 1-14.
- Han J.; Guo X.; Koyama T.; Kawai D.; Zhang J.; Yamaguchi R.; Zhou X.; Motoo Y.; Satoh T.; Yamada S. Zonarol Protected Liver from Methionine-and Choline-Deficient Diet-Induced Nonalcoholic Fatty Liver Disease in a Mouse Model. Nutrients 2021, 13, 3455.
- Jeon S.H.; Jang E.; Park G.; Lee Y.; Jang Y.P.; Lee K.-T.; Inn K.-S.; Lee J.K.; Lee J.-H. Beneficial Activities of Alisma orientale Extract in a Western Diet-Induced Murine Non-Alcoholic Steatohepatitis and Related Fibrosis Model via Regulation of the Hepatic Adiponectin and Farnesoid X Receptor Pathways. Nutrients 2022, 14, 695.
- Le T.N.H.; Choi H.-J.; Jun H.-S. Ethanol Extract of Liriope platyphylla Root Attenuates Non-Alcoholic Fatty Liver Disease in High-Fat Diet-Induced Obese Mice via Regulation of Lipogenesis and Lipid Uptake. Nutrients 2021, 13, 3338.
- Luo Z.; Li M.; Yang Q.; Zhang Y.; Liu F.; Gong L.; Han L.; Wang M. Ferulic acid prevents nonalcoholic fatty liver disease by promoting fatty acid oxidation and energy expenditure in C57bl/6 mice fed a high-fat diet. Nutrients 2022, 14, 2530.
- Songtrai S.; Pratchayasakul W.; Arunsak B.; Chunchai T.; Kongkaew A.; Chattipakorn N.; Chattipakorn S.C.; Kaewsuwan S. Cyclosorus terminans Extract Ameliorates Insulin Resistance and Non-Alcoholic Fatty Liver Disease (NAFLD) in High-Fat Diet (HFD)-Induced Obese Rats. Nutrients 2022, 14, 4895.
- Wang F.; Park J.S.; Ma Y.; Ma H.; Lee Y.J.; Lee G.R.; Yoo H.S.; Hong J.T.; Roh Y.S. Ginseng Saponin Enriched in Rh1 and Rg2 Ameliorates Nonalcoholic Fatty Liver Disease by Inhibiting Inflammasome Activation. Nutrients 2021, 13.
- Zhao H.; Gao X.; Liu Z.; Zhang L.; Fang X.; Sun J.; Zhang Z.; Sun Y. Sodium Alginate Prevents Non-Alcoholic Fatty Liver Disease by Modulating the Gut–Liver Axis in High-Fat Diet-Fed Rats. Nutrients 2022, 14, 4846.
- Zhao W.; Guo M.; Feng J.; Gu Z.; Zhao J.; Zhang H.; Wang G.; Chen W. Myristica fragrans Extract Regulates Gut Microbes and Metabolites to Attenuate Hepatic Inflammation and Lipid Metabolism Disorders via the AhR–FAS and NF-κB Signaling Pathways in Mice with Non-Alcoholic Fatty Liver Disease. Nutrients 2022, 14, 1699.
- Zhao W.; Song F.; Hu D.; Chen H.; Zhai Q.; Lu W.; Zhao J.; Zhang H.; Chen W.; Gu Z. The protective effect of Myristica fragrans Houtt. extracts against obesity and inflammation by regulating free fatty acids metabolism in nonalcoholic fatty liver disease. Nutrients 2020, 12, 2507.
- Zhong Y.; Li Z.; Jin R.; Yao Y.; He S.; Lei M.; Wang X.; Shi C.; Gao L.; Peng X. Diosgenin Ameliorated Type II Diabetes-Associated Nonalcoholic Fatty Liver Disease through Inhibiting De Novo Lipogenesis and Improving Fatty Acid Oxidation and Mitochondrial Function in Rats. Nutrients 2022, 14, 4994.
Overall, we thank the reviewers for their positive comments as well as constructive criticism on our manuscript. We believe the manuscript is now acceptable for publication.
Thank you.
Sincerely,
Woojin Jun, Ph.D
Professor, Chonnam National University

Reviewer 2 Report
In every experiment, CON+SWP are needed. It is not only about cytotoxicity. Other potential impacts need to be excluded, otherwise the author's conclusion cannot be supported.
Author Response

(The authors gave the same response as above.)
